# Genome-Wide Association Study Adjusted for Occupational and Environmental Factors for Bladder Cancer Susceptibility

**DOI:** 10.3390/genes13030448

**Published:** 2022-02-28

**Authors:** Takumi Takeuchi, Mami Hattori-Kato, Yumiko Okuno, Masayoshi Zaitsu, Takeshi Azuma

**Affiliations:** 1Department of Urology, Japan Organization of Occupational Health and Safety, Kanto Rosai Hospital, 1-1 Kizukisumiyoshi-cho, Nakahara-ku, Kawasaki 211-8510, Japan; mami_rainbow_rejoice@yahoo.co.jp (M.H.-K.); yumiyumix@hotmail.com (Y.O.); 2Department of Public Health, Dokkyo Medical University School of Medicine, 880 Kitakobayashi, Mibu, Shimotsuga-gun, Tochigi 321-0293, Japan; m-zaitsu@dokkyomed.ac.jp; 3Department of Urology, Tokyo Metropolitan Tama Medical Center, 2-8-29 Musashidai, Fuchu, Tokyo 183-8524, Japan; tazuma-tky@umin.ac.jp

**Keywords:** bladder cancer, genome-wide, occupation

## Abstract

This study examined the effects of single-nucleotide polymorphisms (SNPs) on the development of bladder cancer, adding longest-held occupational and industrial history as regulators. The genome purified from blood was genotyped, followed by SNP imputation. In the genome-wide association study (GWAS), several patterns of industrial/occupational classifications were added to logistic regression models. The association test between bladder cancer development and the calculated genetic score for each gene region was evaluated (gene-wise analysis). In the GWAS and gene-wise analysis, the gliomedin gene satisfied both suggestive association levels of 10^−5^ in the GWAS and 10^−4^ in the gene-wise analysis for male bladder cancer. The expression of the gliomedin protein in the nucleus of bladder cancer cells decreased in cancers with a tendency to infiltrate and those with strong cell atypia. It is hypothesized that gliomedin is involved in the development of bladder cancer.

## 1. Introduction

There were approximately 83,730 new cases of bladder cancer (64,280 in men and 19,450 in women) and approximately 17,200 deaths from bladder cancer (12,260 in men and 4940 in women) in the United States in 2021. The rates of new bladder cancer and death due to bladder cancer have been decreasing slightly in women in recent years, whereas in men, incidence rates have been decreasing, but death rates are stable [1]. In Japan, 23,230 cases (17,555 in men and 15,675 in women) of bladder cancer were diagnosed in 2018, and the number of deaths from bladder cancer was 8911 (6014 in men and 2897 in women) in 2019 [2].

Muscle invasive bladder cancer requires highly invasive treatments such as radical cystectomy and systemic chemotherapy. In addition, even non-muscle infiltrating bladder cancer often recurs in the bladder and requires multiple treatments. Thus, medical treatment for bladder cancer requires a great deal of time and medical expenses.

It is well known that smoking is a risk factor for developing bladder cancer [3,4]. Regarding alcohol drinking, the American Society of Clinical Oncology stated in 2018 that more than 5% of new cancer cases were due to alcohol consumption [5]. We also reported that alcohol consumption is an independent risk factor for the development of bladder cancer in the Japanese population [6].

Occupational and environmental factors are important, in addition to genetic predisposition for bladder cancer. There was a 45-year observational study in Nordic countries on the association between occupation and the development of bladder cancer. According to this study, occupations with a significantly increased incidence of urothelial cancer, with a standardized incidence ratio of 1.20 or higher, include male waiters, chimney sweeps, hairdressers, assistant nurses, seamen, plumbers, cooks and stewards, beverage workers, female tobacco workers, printers, waiters, chemical process workers, sales agents, hairdressers, mechanics, and administrators [7].

This study examined the effects of single-nucleotide polymorphisms (SNPs) in the germline genome on the development of bladder cancer in Japan, adding occupational and industrial history as regulators.

A genome-wide association study (GWAS) comprehensively searches the entire genome for gene polymorphisms that exhibit significant frequency differences between an unrelated patient population of a specific disease and an unrelated control population.

In genome-wide studies that analyzed the genomes of bladder cancer patients, 57 SNPs that may increase susceptibility to bladder cancer were identified in the GWAS Catalog (Appendix A). In addition, there are many GWAS papers on bladder cancer [8,9,10,11,12,13,14,15,16,17,18,19,20,21,22,23,24,25]. In particular, the NAT2 slow acetylator and GSTM1 null genotype are considered to be potential genetic risk factors for the development of bladder cancer [8]. Polymorphisms in the NAT2 gene were also investigated in Japan, with a risk ratio of 7.80-times [26]. In addition, a relatively large GWAS for Japanese bladder cancer patients was announced in 2015, and although smoking has been examined and adjusted as an environmental factor, occupational factors have not been examined [21]. Therefore, it is important to examine the relationship between bladder cancer in the Japanese population and SNPs by adjusting for the industrial/occupational history, in addition to sex, smoking history, and alcohol drinking history.

## 2. Materials and Methods

The genome was purified using 10 mL of blood mixed with EDTA collected from 352 bladder cancer patients (302 males, 50 females) and 434 control patients (395 males, 39 females) at Japan Organization of Occupational Health and Safety, Kanto Rosai Hospital and Tokyo Metropolitan Tama Medical Center. Control patients did not include those with upper tract urothelial cancer because bladder cancer and upper tract urothelial cancer are considered to be malignant tumors that are anatomically, histologically, and epidemiologically similar.

Occupational and environmental data were obtained from the Inpatient Clinico-Occupational Database of Rosai Hospital Group (ICOD-R), provided by the Japan Organization of Occupational Health and Safety. The ICOD-R includes an occupational history of current and past three jobs, information on smoking, and alcohol habits using interviews and questionnaires completed at the time of admission. Detailed occupational histories were coded with three-digit codes in the Japan Standard Occupational Classification and Japan Standard Industrial Classification corresponding to the International Standard Industrial Classification and International Standard Occupational Classification, respectively [27]. The Japan Standard Occupational Classification is composed of 12 major groups, 74 minor groups, and 329 unit groups [28], whereas the Japan Standard Industrial Classification is composed of 20 divisions, 99 major groups, 530 groups, and 1460 industries [29]. Other clinical data were obtained from electronic medical records. Missing values exist due to omission or lack of description by patients.

### 2.1. A New Classification of Industry/Occupation

To create a new classification, we divided the occupations into four groups: professional, service, management, and blue-collar workers, and further divided the industries into three groups: white-collar industry, blue-collar industry, and service industry. These two kinds of groups were combined into a total of 12 (4 × 3) industry/occupation classes [30,31]. Using this classification, tentatively named the Zaitsu classification, we previously reported an association between occupation and the prognosis of bladder cancer [32].

### 2.2. Clinical and Environmental Factor 

From the clinical data, categorical variables were preliminarily analyzed by Fisher’s exact test between two or multiple groups, and continuous variables were preliminarily analyzed by the Mann–Whitney U test. Furthermore, logistic regression analysis was performed with the development of bladder cancer as the objective variable, whereas age, sex, Brinkman index (BI) classified into four groups (0: BI 0, 1: 1–399, 2: 400–799, 3: 800≤), alcohol consumption history (2 levels, yes or no), and industrial / occupational classifications of the longest-held job for each patient were explanatory variables.

The industrial/occupational classifications added to the logistic regression model here were: (a) industrial classification (20 divisions, Appendix A); (b) the 35 major groups included in industrial classification divisions D (Construction), E (Manufacturing), and H (Transport and postal services); and (c) Occupational classification major groups (12 categories, Appendix A). From the logistic regression models of a, b, and c, the explanatory variables related to industry/occupation were selected by the backward step-wise method using the Akaike information criterion.

### 2.3. Genotyping 

Performed by Riken Genesis Co., Ltd. (Taito-Ku, Tokyo, Japan). Samples were genotyped using the Illumina Infinium Asian Screening Array-24 v1.0 BeadChip, which combines genome-wide coverage of East Asian populations, relevant clinical research content, and scalability for genomic screening. For quality control of samples, we excluded those with (i) a sample call rate < 0.99, (ii) a person with the lowest call rate from the pairs with a proportion IBD (identity-by-descent) > 0.1875, and (iii) outliers from Japanese clusters identified by principal component analysis using the genotyped samples and East Asians in the International Genome Sample Resource [33] (The 1000 Genomes Project Consortium 2015). For quality control of genotypes, we excluded those with a (i) SNP call rate < 0.99 or (ii) *p*-value for the Hardy-Weinberg Disequilibrium test < 0.001.

### 2.4. Imputation 

We utilized SNP imputation for all samples under 1000 Genomes Project Phase 3 as a reference panel [34]. We implemented the pre-phasing by Eagle [35,36] and imputation by Minimac3 [37]. After imputation, we excluded SNPs with an imputation quality of R-square < 0.3.

### 2.5. GWAS 

We conducted 6 GWAS patterns for bladder cancer development by logistic linear models using SNP dosage obtained by SNP imputation and Efficient and Parallelizable Association Container Toolbox (EPACTS) [38]. In the association test, age, sex, smoking history (Brinkman Index, ordered category with 0 < 1 < 2 < 3 levels), alcohol consumption history (2 levels, yes or no), and several patterns of industrial/occupational classifications were added to logistic regression models. Tested industrial/occupational classifications were: (i) 1 variable with 20 levels for industrial classification divisions, (ii) selected industrial classification division(s) from 20 variables with 2 levels (yes or no) by the backward step-wise method in a logistic regression model without SNP dosage, (iii) 1 variable with 12 levels for occupational classification major groups, (iv) selected occupational classification major group(s) from 12 variables with 2 levels (yes or no) by the backward step-wise method in a logistic regression model without SNP dosage, (v) selected industrial classification major groups from 35 variables with 2 levels (yes or no) by the backward step-wise method in a logistic regression model without SNP dosage, and (vi) the Zaitsu classification. We also used only male samples for GWAS, taking into account sex differences in some occupations. We did not conduct GWAS using only females due to the small number of cases. We set the genome-wide significance level for our study at *p* = 5 × 10^−8^ and suggestive association level at *p* = 10^−5^ [39]. 

### 2.6. Gene-Wise Analysis 

For SNPs contained within 50 bp upstream and downstream of the gene regions defined in Ref Gene [40], we calculated the genetic score (GS) [41] as described below, and the association test between bladder cancer development and GS for each gene region was evaluated by the Burden test and SKAT-O test using EPACTS (version 3.2.6) (University of Michigan, Ann Arbor, MI, USA). In our study, we performed gene-wise analysis for 20,865 regions. We set the genome-wide significance level for our study at *p* = 2.4 × 10^−6^ (=0.05/20,865) and suggestive association level at *p* = 10^−4^. Adjusting factors in GWAS were also included in the gene-wise analysis. GWAS and gene-wise analysis were performed by StaGen Co., Ltd. (Taito-ku, Tokyo, Japan 111-0051).
(1)$$GS_{i}=\overset{M}{\underset{j=1}{\sum }}x_{ij}\beta_{j}/M$$

Here, the *GS_i_* of an individual patient is equal to the weighted sum of the individual’s genotypes, *x_j_* (0, 1, 2), at SNPs in *gene_i_*. Weights (*β_j_*) are calculated by EPACTS and M is the number of SNPs in *gene_i_*.

### 2.7. Immunohistochemistry 

The expression of gliomedin protein was examined by tissue immunostaining using paraffin-embedded bladder tumor tissue removed by transurethral resection of the bladder tumor. The antibody used was anti-GLDN (gliomedin) polyclonal antibody (26185-1-AP, Proteintech, Rosemont, IL, USA). Two independent pathologists evaluated histological staining by the immunoreactive score [42] and individual scores were analyzed after averaging.

### 2.8. Study Approval 

The Ethical Committee of the Japan Organization of Occupational Health and Safety approved the experiments (2018-2). All experiments were performed in accordance with relevant guidelines and regulations, including any relevant details. Written informed consent was received from patients prior to inclusion in the study.

## 3. Results

### 3.1. Clinical and Environmental Factors 

The age of the bladder cancer patients included was slightly lower than the control patients for men and higher for women (Table 1). Malignant tumors other than urothelial cancer were observed in 13.9% of men and 18.0% of women in the bladder cancer group, and 72.2% of men and 59.0% of women in the control group. In the male control group, 46.6% had prostate cancer and 11.1% had renal cell carcinoma (Appendix A).

In terms of smoking history, a high Brinkman index classified into four stages and the development of male bladder cancer were slightly related. The Brinkman index 2–3 group had more male bladder cancer than the Brinkman index 0–1 group (Table 1). As for alcohol consumption history, bladder cancer patients drank slightly less alcohol than control patients overall (Table 1). The overall distributions of the divisions of industrial classification (Table 2), occupational classification major groups (Table 3), and groups in the Zaitsu classification (Table 4) were not significantly different from controls in male, female, and all bladder cancer patients. Looking at the individual divisions of industrial classification, bladder cancer was less frequent in division G and more frequent in division S in male cases and all cases (Table 2). In addition, in the individual major groups of occupational classification, bladder cancer was significantly more common in the major group F in male cases and all cases (Table 3).

In the selection of explanatory variables concerning industry/occupation by the backward step-wise method for logistic regression models, industrial classification divisions G and S, and occupational classification major group F, were selected in male cases from all divisions and from all major groups, respectively, whereas in all cases including both men and women, industrial classification divisions G, L, and S, and occupational classification major group F remained. 

In addition, from the major groups in industrial classification divisions D, E, and H, “Manufacture of general-purpose machinery”, “Miscellaneous manufacturing industries”, “Construction work by specialist contractor”, “Equipment installation work”, and “Railway transport” were selected as explanatory variables in male cases, whereas “Miscellaneous manufacturing industries”, “Construction work by specialist contractor”, and “Railway transport” remained in all cases. The overall distribution of bladder cancer cases in these industrial major groups was not different from that of the controls (Table 5).

### 3.2. Sample QC 

For 789 genotyped samples of 830 samples, 21 were excluded in which the sample call rate was <0.99, the proportion IBD was >0.1875, and outliers from Japanese clusters identified by principal component analysis (Appendix A). In our study, we used 766 samples in GWAS (Appendix A). 

### 3.3. SNP QC for Imputation

We selected SNPs to be used for SNP imputation. The number of SNPs loaded on the chip was 659,184 and the number of SNPs genotyped was 657,060. In addition, there were 641,043 SNPs with a definite chromosomal location, 395,708 SNPs with a call rate of 99% or higher, a *p*-value of 0.0001 for the Hardy–Weinberg law of equilibrium, and a minor allele frequency (MAF) of 1% or higher. SNP imputation was performed using 395,708 SNPs (Appendix A).

### 3.4. Imputation

The number of SNPs able to be imputed was 47,109,297. Of the 47,109,297 SNPs, 11,175,945 with an R-squared value greater than 0.3 were used in the GWAS (Appendix A).

### 3.5. Results of GWAS and Gene-Wise Analysis

No SNPs satisfying the genome-wide significance level 5 × 10^−8^ were detected in GWAS in all cases or in male cases. A Manhattan plot of the genome-wide association test for each analysis pattern is shown in Figure 1. In the gene-wise analysis, no genes satisfying the genome-wide significance level 2.4 × 10^−6^ were detected in all cases or in male cases. In GWAS and gene-wise analysis, the gliomedin (*GLDN*) gene located at 15q21.2 satisfied both a suggestive association level of 10^−5^ in GWAS and suggestive association level of 10^−4^ in gene-wise analysis (Table 6 and Figure 2).

In the Manhattan plot of male bladder cancer cases, there were peaks satisfying a suggestive level of *p* < 10^−5^ between LINC00922 and *CDH5* in chromosome 16, and between LINC00473 and *PDE10A* in chromosome 6, but they were outside the genetic regions in the regional plots of GWAS results (Appendix A).

In all bladder cancer cases, Manhattan plot peaks satisfying *p* < 10^−5^ were observed near *WNT2B* in chromosome 1 and *XYLB* of chromosome 3 (Appendix A), but they did not satisfy the suggestive association level of *p* < 10^−4^ in gene-wise analysis.

### 3.6. Expression of Gliomedin Protein in Bladder Cancer Tissues

The expression of the gliomedin protein (Figure 3) in the nucleus of bladder cancer cells was lower in cancers with a tendency to infiltrate and those with strong cell atypia, as shown in Table 7. The expression of the gliomedin protein in the cytoplasm of cancer cells and in the nucleus of stromal cells was not associated with the degree of cancer infiltration or cancer cell atypia.

## 4. Discussion

Kawasaki City, where Kanto Rosai Hospital is located, is a traditional heavy industry area adjacent to the Tokyo Metropolitan area. Fuchu City, where Tokyo Metropolitan Tama Medical Center is located, is a commercial and residential area on the outskirts of Tokyo. Therefore, workers engaged in the primary sector of industry and mining industry were limited in this study.

The reason why the industrial classification divisions G, L, and S, and the occupational classification major group F were particularly adopted as the adjusting factors in certain patterns of GWAS analysis is that these factors were selected by the backward step-wise method in the analysis of the patient background. In addition, the adjusting factors were selected from the industrial classification major groups included in the industrial classification divisions D, E, and H for men because a relatively large number of cases were included in these three divisions.

The occupations vulnerable to bladder cancer in previous reports were fairly specific and limited. Compared with these, this study mainly used the relatively rough classification of industry/occupation such as industrial classification divisions, major groups, and occupational classification major groups. Therefore, the purpose of this study was not to examine the relationship between SNPs and specific environmentally exposed substances, such as nicotine and aromatic amines, but rather to incorporate the contribution of broader industrial/occupational environmental factors, such as stress stimulation and work environment, into the development of bladder cancer as adjustment factors. Under these conditions, the gliomedin gene was detected in this study by GWAS and gene-wise analysis as a gene that may be associated with the development of bladder cancer in males.

GWAS is widely performed to replicate obtained results with other datasets. However, detailed recording of occupational/industrial history, such as ICOD-R, is not comprehensively enforced in Japan, making it difficult to replicate GWAS with occupational/industrial history as an adjusting factor. Therefore, in this study, in addition to GWAS to verify one SNP, the results were supported by performing gene-wise analysis to examine the association between a given pathological condition and a certain gene as a whole.

Kaneko et al. used ICOD-R occupational classification major groups to demonstrate that occupations with high physical activity reduced the risk of cancer [27]. They also compared the categories included in the manufacturing industry division (Division E) of ICOD-R and noted that the incidence of ureter cancer in the electronics category is higher than that in the food manufacturing category [43]. Therefore, adding the industrial/occupational classification to the adjusting factors of GWAS, even if it is relatively rough, is considered to be meaningful in examining the development of cancer.

The control group in this analysis included several malignant tumor diseases other than urothelial cancer. The inclusion of many cases of other malignancies in the control group of GWAS for bladder cancer is controversial. It is thought that pathways common to malignancies in general are less likely to be detected, but on the other hand, it may be more effective in order for pathways specific to bladder cancer to emerge.

The gliomedin gene encodes a protein containing olfactomedin-like and collagen-like domains. The gliomedin protein, which is present in both transmembrane and secretory forms, promotes the formation of the Node of Ranvier in the peripheral nervous system [44]. Mutations in the gliomedin gene cause lethal congenital contracture syndrome [45]. Autoantibodies to the gliomedin protein have also been identified in patients with multifocal motor neuropathy serotypes [46]. An important paralog of the gliomedin gene is olfactomedin protein family [47].

The expression of gliomedin mRNA and protein is found in the nuclei of many types of cancer cells, including urothelial cancer [48]. The early deregulation of gliomedin during liver tumorigenesis was previously reported [49]. The gliomedin paralog, olfactomedin 4, is a glycoprotein with an olfactomedin-domain, which is involved in numerous intracellular signaling pathways, including NF-κB, and is associated with innate immunity. Furthermore, olfactomedin 4 suppresses the development and progression of cancer [50,51], and tumorigenesis is observed in olfactomedin4 deficient mice [52]. Thus, it is hypothesized that gliomedin is also involved in the development of bladder cancer by a mechanism similar to that of olfactomedin 4 in innate immunity and oncogenesis in a certain environment.

As the expression of the gliomedin protein in the nucleus of cancer cells is decreased in bladder cancer with strong nuclear atypia and infiltration tendency, it is speculated that gliomedin may act as a tumor suppressor factor in bladder cancer. As GWAS suggested a relationship between the gliomedin gene and the development of bladder cancer in men, further studies are required. 

## 5. Conclusions

In conclusion, the gliomedin gene was suggested to be related to the development of male bladder cancer by adding longest-held occupational and industrial history as regulators in the GWAS and gene-wise analysis. In addition, the expression of the gliomedin protein in the nucleus of bladder cancer cells was lower in cancers with a tendency to infiltrate and those with strong cell atypia.

## Figures and Tables

**Figure 1 genes-13-00448-f001:**
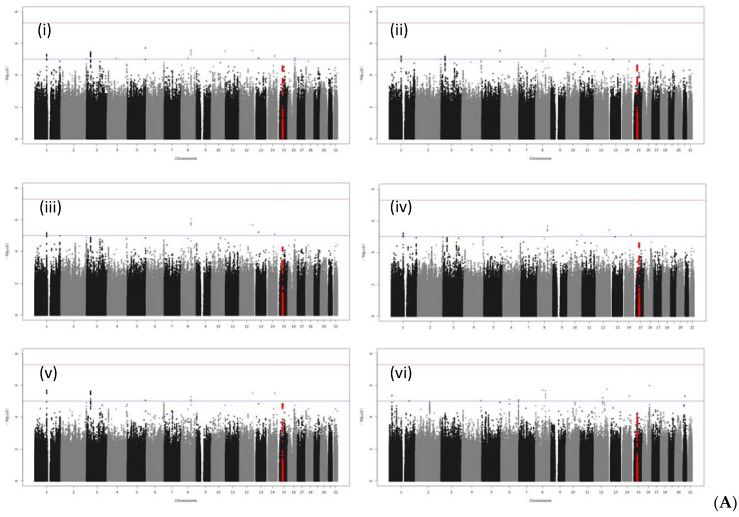
Manhattan plots of GWAS: (**A**) All bladder cancer; (**B**) male bladder cancer. Industrial/occupational factors added in GWAS: (i) 1 variable with 20 levels for industrial classification divisions; (ii) selected industrial classification divisions G and S for male bladder cancer, with divisions G, L, and S for all bladder cancer; (iii) 1 variable with 12 levels for occupational classification major groups; (iv) selected occupational classification major group F; (v) selected industrial classification major groups in D, E, and H, i.e., “Manufacture of general-purpose machinery”, “Miscellaneous manufacturing industries”, “Construction work by specialist contractor”, “Equipment installation work”, and “Railway transport” for male bladder cancer, and “Miscellaneous manufacturing industries”, “Construction work by specialist contractor”, and “Railway transport” for all bladder cancer; and (vi) the Zaitsu classification, The SNPs in the *GDLN* region are plotted in red.

**Figure 2 genes-13-00448-f002:**
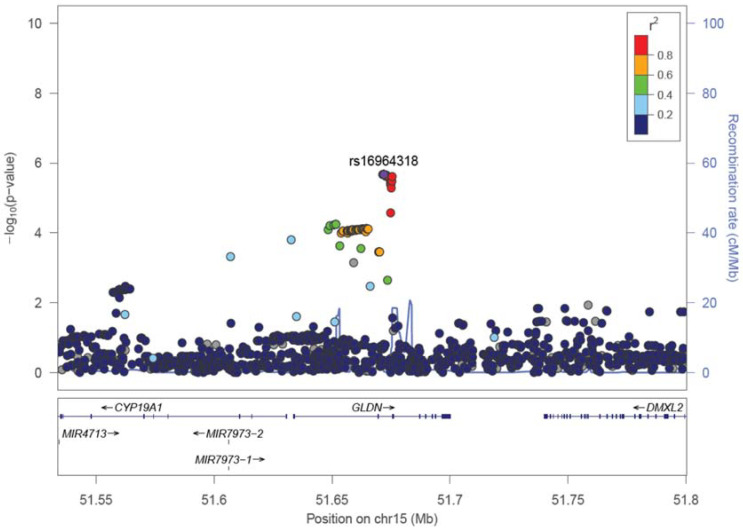
A representative regional plot of *GLDN* region. Added industrial/occupational factors were selected among industrial classification major groups in D, E, and H, i.e., “Manufacture of general-purpose machinery”, “Miscellaneous manufacturing industries”, “Construction work by specialist contractor”, “Equipment installation work”, and “Railway transport” for male bladder cancer.

**Figure 3 genes-13-00448-f003:**
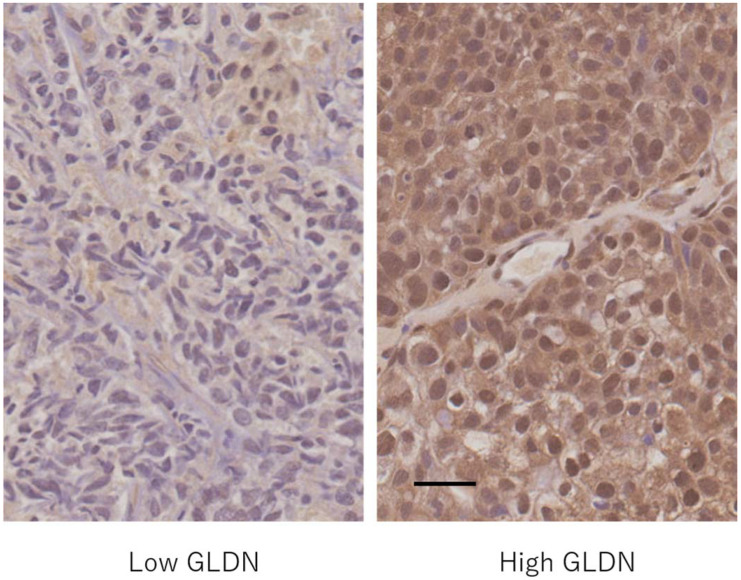
Gliomedin (GLDN) protein expression in bladder cancer. Left: a case with low gliomedin expression in both the nucleus and cytoplasm of cancer cells; Right: a case with high gliomedin expression in both the nucleus and cytoplasm of cancer cells. Scale bar = 0.5 μm.

**Table 1 genes-13-00448-t001:** Clinical and environmental factors.

	Male		Female		All	
	Bladder Ca	Control	Bladder Ca	Control	Bladder Ca	Control
Age (yrs)	*n* = 300	*n* = 395	*n* = 50	*n* = 39	*n* = 350	*n* = 434
	71.4 ± 10.3	72.8 ± 11.8	72.0 ± 10.1	64.6 ± 12.1	71.5 ± 10.2	72.1 ± 12.1
*p*-value	0.0079		0.0030		0.0850	
Brinkman index (BI)	*n* = 282	*n* = 298	*n* = 48	*n* = 27	*n* = 330	*n* = 325
0	27.7%	29.9	83.3%	81.5	35.8%	35.0
1	13.5	19.1	6.3	11.1	12.4	15.4
2	27.3	24.2	2.1	3.7	23.6	23.1
3	31.6	26.8	8.3	3.7	28.2	24.9
*p*-value for 2 × 4 columns	0.0952		0.7276		0.1073	
*p*-value for BI 0 vs. BI 1–3	0.4621		1.0000		0.8696	
*p*-value for BI 0–1 vs. BI 2–3	0.0288		1.0000		0.1569	
Alcohol consumption	*n* = 281	*n* = 291	*n* = 47	*n* = 27	*n* = 328	*n* = 318
0	23.5%	21.3	63.8%	63.0	29.3%	23.0
1	76.5	78.7	36.2	37.0	70.7	77.0
*p*-value	0.2222		1.0000		0.0736	

Ages were analyzed by the Mann–Whitney U test, whereas the Brinkman index and alcohol consumption were analyzed by Fisher’s exact test between two or multiple groups.

**Table 2 genes-13-00448-t002:** Distribution of industrial classification divisions.

Industrial Classification	A	B	C	D	E	F	G	H	I	J	K	L	M	N	O	P	Q	R	S	T
Male																				
Bladder Ca (*n* = 236)	0.8	0.0	0.0	11.0	25.4	2.5	1.3 *	9.3	11.4	3.4	1.3	5.9	3.4	2.1	2.1	1.7	0.8	3.0	5.9 *	8.5
Control (*n* = 289)	1.7	0.0	0.0	11.4	26.3	1.7	5.2 *	8.7	10.0	2.8	2.1	3.5	3.8	1.4	1.4	2.4	0.3	3.8	1.7 *	11.8
Female																				
Bladder Ca (*n* = 41)	2.4	0.0	0.0	0.0	14.6	0.0	0.0	2.4	12.2	4.9	0.0	2.4	2.4	0.0	0.0	2.4	0.0	4.9	0.0	51.2
Control (*n* = 27)	0.0	0.0	0.0	0.0	18.5	0.0	0.0	0.0	7.4	0.0	3.7	0.0	7.4	0.0	11.1	3.7	0.0	3.7	0.0	44.4
All																				
Bladder Ca (*n* = 277)	1.1	0.0	0.0	9.4	23.8	2.2	1.1 *	8.3	11.6	3.6	1.1	5.4	3.2	1.8	1.8	1.8	0.7	3.2	5.1 *	14.8
Control (*n* = 316)	1.6	0.0	0.0	10.4	25.6	1.6	4.7 *	7.9	9.8	2.5	2.2	3.2	4.1	1.3	2.2	2.5	0.3	3.8	1.6 *	14.6

See Appendix A for details of industrial classification divisions. *p*-values analyzed among multiple groups by Fisher’s exact test were 0.2929, 4873, and 0.3358 for males, females, and all, respectively. Asterisks indicate *p*-values less than 0.05 analyzed by Fisher’s exact test in individual 2 × 2 groups. Classifications G and S in males and G, L, and S in all were selected by the backward step-wise method.

**Table 3 genes-13-00448-t003:** Distribution of occupational classification major groups.

Occupational Classification	A	B	C	D	E	F	G	H	I	J	K	L
Male												
Bladder Ca (*n* = 246)	4.2	19.9	15.3	15.7	4.2	3.8 *	0.8	12.7	5.5	6.8	2.5	4.2
Control (*n* = 291)	5.2	18.0	15.6	15.6	5.9	0.3 *	2.4	12.5	4.8	6.2	1.7	5.2
Female												
Bladder Ca (*n* = 20)	0.0	2.4	12.2	14.6	4.9	0.0	0.0	9.8	0.0	0.0	4.9	51.2
Control (*n* = 15)	0.0	14.8	14.8	11.1	14.8	0.0	0.0	3.7	0.0	0.0	0.0	44.4
All												
Bladder Ca (*n* = 266)	3.6	17.3	14.8	15.5	4.3	3.2 *	0.7	12.3	4.7	5.8	2.9	14.8
Control (*n* = 306)	4.7	17.7	15.5	15.2	6.3	0.3 *	2.2	11.7	4.4	5.7	1.6	14.6

See Appendix A for details of occupational classification major groups. *p*-values analyzed among multiple groups by Fisher’s exact test were 0.2209, 0.9430, and 0.2979 for males, females, and all, respectively. Asterisks indicate *p*-values less than 0.05 analyzed by Fisher’s exact test in individual 2 × 2 groups. Classification F in males and all was selected by the backward step-wise method.

**Table 4 genes-13-00448-t004:** Distribution of groups in the Zaitsu classification.

	Blue-Collar Industry				Service Industry				White-Collar Industry			
	Blue-Collar Worker	Service Worker	Professional	Manager	Blue-Collar Worker	Service Worker	Professional	Manager	Blue-Collar Worker	Service Worker	Professional	Manager
Male												
Bladder Ca (*n* = 216)	21.8	19.4	10.2	2.3	5.1 *	14.8	1.4	1.4	3.2	9.3	10.2	0.9
Control (*n* = 255)	23.5	19.2	9.8	3.9	1.2 *	18.4	0.4	2.0	0.8	10.2	10.6	0.0
Female												
Bladder Ca (*n* = 20)	25.0	15.0	0.0	0.0	5.0	35.0	0.0	0.0	0.0	10.0	10.0	0.0
Control (*n* = 35)	17.1	20.0	0.0	0.0	2.9	31.4	2.9	0.0	0.0	8.6	17.1	0.0
All												
Bladder Ca (*n* = 236)	22.0	19.1	9.3	2.1	5.1 *	16.5	1.3	1.3	3.0	9.3	10.2	0.8
Control (*n* = 290)	22.8	19.3	8.6	3.4	1.4 *	20.0	0.7	1.7	0.7	10.0	11.4	0.0

*p*-values analyzed among multiple groups by Fisher’s exact test were 0.1424, 0.4563, and 0.1999 for males, females, and all, respectively. Asterisks indicate *p*-values less than 0.05 analyzed by Fisher’s exact test in individual 2 × 2 groups.

**Table 5 genes-13-00448-t005:** Distribution of industrial classification major groups in divisions D, E, and H in males.

Major Group	Bladder Ca_Male (%)	Control_Male (%)
Manufacture of food	1.9	3.7
Manufacture of beverages, tobacco, and feed	0.0	0.7
Manufacture of textile products	0.9	1.5
Manufacture of lumber and wood products, except furniture	0.0	0.7
Manufacture of furniture and fixtures	0.9	0.0
Manufacture of pulp, paper, and paper products	0.0	0.7
Printing and allied industries	1.9	0.0
Manufacture of chemical and allied products	4.6	3.0
Manufacture of petroleum and coal products	0.0	0.0
Manufacture of plastic products, except otherwise classified	3.7	0.0
Manufacture of rubber products	0.0	0.0
Manufacture of leather tanning, leather products and fur skins	0.9	0.0
Manufacture of ceramic, stone, and clay products	0.0	3.0
Manufacture of iron and steel	3.7	2.2
Manufacture of non-ferrous metals and products	1.9	1.5
Manufacture of fabricated metal products	6.5	9.0
*Manufacture of general-purpose machinery*	0.9	3.7
Manufacture of production machinery	3.7	1.5
Manufacture of business-oriented machinery	2.8	5.2
Electronic parts, devices, and electronic circuits	4.6	4.5
Manufacture of electrical machinery, equipment, and supplies	2.8	2.2
Manufacture of information and communication electronics equipment	4.6	6.0
Manufacture of transportation equipment	5.6	6.7
*Miscellaneous manufacturing industries*	3.7	0.7
Construction work, general including public and private construction work	8.3	9.0
*Construction work by specialist contractor, except equipment installation work*	11.1	5.2
*Equipment installation work*	4.6	10.4
*Railway transport*	4.6	1.5
Road passenger transport	6.5	6.7
Road freight transport	5.6	8.2
Water transport	0.9	0.0
Air transport	0.9	0.0
Warehousing	0.9	1.5
Services incidental to transport	0.9	0.7
Postal services, including mail delivery	0.0	0.0

The *p*-value analyzed among multiple groups by Fisher’s exact test was 0.2649 for males. Groups in italics were selected by the backward step-wise method.

**Table 6 genes-13-00448-t006:** Results of GWAS and gene-wise analysis for bladder cancer. Results that satisfied both *p* < 10^−5^ by GWAS and *p* < 10^−4^ by gene-wise analysis were selected.

	Annotation				Allele		Frequency		GWAS			Gene-Wise		
Analysis	Chr	Gene	rsID	BP	Ref	Alt	Case	Control	Beta	SE	*p* Value	Beta	SE	*p* Value
(i)	15	*GLDN*	rs10162956	51673125	C	T	0.13	0.06	1.26	0.28	6.37 × 10^−6^	0.1	0.02	3.13 × 10^−5^
(ii)	15	*GLDN*	rs10162956	51673125	C	T	0.13	0.06	1.26	0.27	4.39 × 10^−6^	0.1	0.02	1.84 × 10^−5^
(iii)	15	*GLDN*	rs28619121	51671391	C	T	0.13	0.06	1.22	0.27	8.70 × 10^−6^	0.1	0.02	3.80 × 10^−5^
(iv)	15	*GLDN*	rs16964318	51671920	T	C	0.13	0.06	1.23	0.27	5.78 × 10^−6^	0.1	0.02	2.35 × 10^−5^
(v)	15	*GLDN*	rs16964318	51671920	T	C	0.13	0.06	1.31	0.28	2.17 × 10^−6^	0.1	0.02	1.57 × 10^−5^

Analysis: industrial/occupational factors added in GWAS and gene-wise analysis in male bladder cancer: (i) 1 variable with 20 levels for industrial classification divisions; (ii) selected industrial classification divisions G and S for male bladder cancer; (iii) 1 variable with 12 levels for occupational classification major groups; (iv) selected occupational classification major group F; (v) selected industrial classification major groups in D, E, and H, i.e., “Manufacture of general-purpose machinery”, “Miscellaneous manufacturing industries”, “Construction work by specialist contractor”, “Equipment installation work”, and “Railway transport”, Chr: chromosome, BP: base pair position, Ref: reference allele, Alt: alternative allele, SE: standard error.

**Table 7 genes-13-00448-t007:** Expression of gliomedin protein in bladder cancer tissues.

	Cancer Cell Nucleus	Cancer Cell Cytoplasm	Stromal Cell Nucleus
With muscle invasion (*n* = 13)	4.42 ± 2.01 *	6.50 ± 2.85	3.73 ± 1.58
Without muscle invasion (*n* = 54)	5.84 ± 2.37 *	6.24 ± 2.89	4.16 ± 1.76
With submucosal invasion (*n* = 30)	5.02 ± 1.77 **	6.52 ± 2.46	3.95 ± 1.53
Without submucosal invasion (*n* = 37)	5.99 ± 2.68 **	6.11 ± 3.16	4.18 ± 1.87
High-grade cancer (*n* = 25)	4.78 ± 1.69 #	6.82 ± 2.62	3.98 ± 1.31
Low-grade cancer (*n* = 40)	6.08 ± 2.60 #	6.09 ± 2.97	4.19 ± 1.96

*: *p* = 0.0395, **: *p* = 0.0827, #: *p* = 0.0186 by the Welch’s Two Sample *t*-test, values: mean ± SD, high grade: grade 3; low grade: grades 1 and 2.

## Data Availability

Not applicable.

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
