# Peer review of "Genome-Wide Association Study Adjusted for Occupational and Environmental Factors for Bladder Cancer Susceptibility"

_genes, 2022, doi:10.3390/genes13030448_

Round 1

Reviewer 1 Report

The study is well designed and presented. Minor English spell check required. It will be better if the authors provide more representative images of Gliomedin IHC. Scale bars must be included in the IHC images. 

Author Response

The study is well designed and presented. Minor English spell check required. It will be better if the authors provide more representative images of Gliomedin IHC. Scale bars must be included in the IHC images.

Thanks for comments. I requested English-proofreading, so I will attach the certificate at the bottom of this letter. I have reviewed many of the photos again, but the photos included in the first draft are more or less representative. Therefore, I would like to keep the previous photos as a representative. Scale bar is depicted in the photo.

Reviewer 2 Report

The authors Takumi Takeuchi et al in here examined the effects of single nucleotide polymorphisms (SNPs) on the development of bladder cancer, adding longest-held occupational and industrial history as regulators. The genome purified from blood was genotyped, followed by SNP imputation. And the expression of gliomedin protein in the nucleus of bladder cancer cells decreased in cancers with a tendency to infiltrate and those with strong cell atypia. It is hypothesized that gliomedin is involved in the development of bladder cancer. In view of relationship between SNPs and the development of bladder cancer are poorly understood, which represents a significant knowledge gap. Such knowledge is fundamental to the understanding of bladder cancer diseases. Addressing this critical knowledge gap will facilitate the bladder cancer’s long-term goal of diagnosis and treatment. So, I highly recommend to publish it after some necessary revision.

  1. Compared to preciously published papers such as Ma, Zhicheng, et al. "Systematic evaluation of bladder cancer risk‐associated single‐nucleotide polymorphisms in a Chinese population." Molecular carcinogenesis52.11 (2013): 916-921.; Wang, Ping, et al. "Genetic score of multiple risk‐associated single nucleotide polymorphisms is a marker for genetic susceptibility to bladder cancer." Genes, Chromosomes and Cancer53.1 (2014): 98-105.; Jordahl, Kristina M., et al. "Mediation by differential DNA methylation of known associations between single nucleotide polymorphisms and bladder cancer risk." BMC Medical Genetics 21.1 (2020): 1-9.; Laytragoon Lewin, Nongnit, et al. "Influence of single nucleotide polymorphisms among cigarette smoking and non-smoking patients with coronary artery disease, urinary bladder cancer and lung cancer." Plos one 16.1 (2021): e0243084; the principle and innovation of this paper should be explained more in details.

  1. There are many abbreviations in use. Please show the full names before use them.

  1. There are some minor errors such as there are some grammar errors which should be revised carefully.

  1. Adjust the format of reference.

Author Response

The principle and innovation of this paper should be explained more in details.

I would like to re-emphasize “Therefore, it is important to examine the relationship between bladder cancer in the Japanese population and SNPs by adjusting for the industrial/occupational history, in addition to sex, smoking history, and alcohol drinking history.” described in the introduction, as both genetic background and environmental factors, including occupational history, are thought to contribute to the development of bladder cancer.

There are many abbreviations in use. Please show the full names before use them.

When the abbreviation first appeared in the paper, I tried to spell out them.

There are some minor errors such as there are some grammar errors which should be revised carefully.

I requested English-proofreading, so I will attach the certificate to the revised cover letter.

Adjust the format of reference.

I modified the Reference. The ACS style lists all authors. GWAS papers often include a large number of authors, and then, the reference part becomes very long. I think it would be better to adopt another citation style to omit author names.